# Analogy of the Reactions of Aromatic and Aliphatic π-Electrophiles with Nucleophiles

**DOI:** 10.3390/molecules28104015

**Published:** 2023-05-11

**Authors:** Michał Barbasiewicz, Michał Fedoryński, Rafał Loska, Mieczysław Mąkosza

**Affiliations:** 1Faculty of Chemistry, University of Warsaw, 02-093 Warsaw, Poland; barbasiewicz@chem.uw.edu.pl; 2Faculty of Chemistry, Warsaw University of Technology, Noakowskiego 3, 00-664 Warsaw, Poland; michal.fedorynski@pw.edu.pl; 3Institute of Organic Chemistry, Polish Academy of Sciences, Kasprzaka 44/52, 01-224 Warsaw, Poland

**Keywords:** nucleophilic substitution, nucleophilic addition, nucleophiles, electrophiles, elimination, nitroarenes, reaction mechanisms

## Abstract

The aim of this essay is to disclose the similarity of a great variety of reactions that proceed between nucleophiles and π-electrophiles—both aromatic and aliphatic. These reactions proceed via initial reversible addition, followed by a variety of transformations that are common for the adducts of both aliphatic and aromatic electrophiles. We hope that understanding of this analogy should help to expand the scope of the known reactions and inspire the search for new reactions that were overlooked.

## 1. Introduction

This essay aims to disclose the similarity of a great variety of reactions that proceed between nucleophiles and aliphatic and aromatic π-electrophiles. There is a common belief that these reactions are mechanistically different, which we believe to be incorrect. Therefore, we hope that understanding and acceptance of this analogy should help to expand the scope of the known reactions and inspire the search for new reactions that were so far overlooked.

Although there is a great variety of **aliphatic** π-electrophiles (carbonyl compounds, imines, iminium ions, nitriles, Michael acceptors, etc.) and **aromatic** π-electrophiles with electrophilic centers in the ring: nitroarenes, azines, etc., as well as a great, practically unlimited variety of nucleophiles; therefore, numerous reactions that proceed between them—there are just a few fundamental initial processes. All these reactions proceed via the initial reversible addition of nucleophiles to the electrophilic center of an electrophilic partner to form σ adducts that are subsequently converted into the final products in a variety of ways. The addition to aliphatic electrophiles results in the formation of a new σ bond and conversion of the π bond into a σ bond, which as a rule is energetically favorable, with the formation of adducts that are energetically stable intermediates. Due to stability, upon protonation, they can form final products, or they can react further in a great variety of ways. On the other hand, the addition of nucleophiles to electron-deficient arenes results in dearomatization, thus it is energetically unfavorable so, as a rule, the produced σ adducts are unstable, short-lived intermediates. These σ adducts tend to recover aromaticity via dissociation or fast further transformations, thus the number of different final reactions appears to be limited. However, recent literature reports indicate that adducts of nucleophiles to aromatic electrophiles can enter a variety of reactions, and thus they display reactivity identical to their aliphatic counterparts [1,2,3].

The second difference between aliphatic and aromatic π-electrophiles is that aromatic π-electrophiles usually contain more than one electrophilic center that is capable of adding nucleophilic agents, and the addition can result in the formation of isomeric adducts. In turn, aliphatic π-electrophiles usually react selectively, giving single, predictable intermediates. However, the difference also seems to be apparent, as it arises from the structure of individual substrates, and varies depending on substitution or length of the conjugation pathway.

These observations should inspire the search for new processes based on the analogies between aromatic and aliphatic π-electrophiles. In this essay, the similarity of reactions of nucleophiles with aromatic and aliphatic π-electrophiles will be presented consecutively, and for the sake of clarity, electrophiles and nucleophiles with or without leaving groups will be discussed separately. We believe that the clear-cut analogy between aromatic and aliphatic π-electrophiles should inspire the search for new synthetic transformations, which were overlooked, due to the simplified picture of their reactivity, presented in chemical textbooks.

## 2. Addition of Nucleophiles to π-Electrophiles Containing Leaving Groups at Electrophilic Center

### 2.1. Addition of Nucleophiles without Leaving Groups

The most common and studied reaction between nucleophiles and aromatic electrophiles is substitution of halogens and other leaving groups in nitroarenes. This reaction, known for more than 150 years, proceeds via the addition of nucleophiles at positions *ortho* or *para* to the nitro group occupied by a leaving group X to form σ^X^ adducts. As was mentioned earlier, the addition is connected with dearomatization, thus the adducts undergo fast rearomatization via the departure of X^−^ to form products of substitution, S_N_Ar. As a consequence, in this two-step process, the addition is the slower, rate-limiting step, therefore usually the substitution proceeds faster when X = F than X = Cl [4].

In some cases, when X = Cl, the departure of Cl^−^ is so fast that the process has a synchronous character [5,6]. It is therefore a peculiar situation that the two-step substitution of fluorine proceeds faster than the synchronous substitution of chlorine.

Substitution of leaving groups in electron-deficient arenes (nitroarenes, azines, etc.) proceeds with various C, N, O, S, etc. nucleophiles. The most common leaving groups are halogens, but alkoxy and aryloxy groups can also behave as leaving groups (Figure 1). The substitution of fluoride in 2-fluoropyridine is a versatile way of preparing pyridine derivatives [7].

As previously mentioned, in *p*- and *o*-halonitrobenzenes and analogues there are also positions occupied by hydrogen available for the addition of nucleophiles. It should be stressed that addition of nucleophiles at these positions is, as a rule, faster than at positions occupied by halogen, but the σ^H^ adducts have no direct way for further conversion; hence, they usually dissociated. Nevertheless, when such a possibility exists, further conversion of the σ^H^ adducts results in nucleophilic substitution of hydrogen. The general picture is presented in Figure 2 [4,8].

There is a great variety of aliphatic π-electrophiles containing leaving groups at the electrophilic centers. The simplest examples of such electrophiles are acyl chlorides (halides), esters of carboxylic acids, imidoyl chlorides, etc., as well as electron-deficient alkenes containing leaving groups: β-halo or β-alkoxyvinyl ketones, nitriles, esters, etc. (Figure 3).

Numerous common reactions such as acylation of carbanions by acyl chlorides and esters, synthesis of esters via acylation of alcohols and transesterification, synthesis of amides, hydrazides, hydroxamic acids, etc. proceed via the addition of nucleophiles to carbonyl and amino groups substituted with a leaving group, followed by elimination of the leaving group (Figure 4). Many names and common reactions, e.g., Claisen condensation, transesterification, etc., belong to this category. It appears difficult to see the analogy between the Claisen condensation and S_N_Ar reaction, but indeed these reactions proceed similarly as follows: addition of nucleophile followed by elimination of a leaving group from the addition center.

The reactions of nucleophiles with π-electrophiles containing leaving groups at the electrophilic centers of electron deficient alkenes are less common. Nevertheless, there are many examples of substitution of β-halogens or β-alkoxy groups in vinyl nitriles, ketones, or sulfones that proceed via an addition–elimination mechanism analogous to S_N_Ar [9,10]. Similar to the S_N_Ar, addition is usually the rate-determining step, hence for instance substitution of fluorine is faster than chlorine (Figure 5). Selected synthetic examples of nucleophilic substitution in aliphatic systems are presented in Figure 6 [11,12,13].

Similar to the case of aromatic electrophiles there are two main mechanistic variants of substituting chlorine in β-chlorovinyl ketones through two-step processes in which the adducts are intermediates or synchronous reaction when the dissociation is faster than addition and the adducts are transition states [9,10]. Occasionally, reactions other than elimination may be observed following the addition of nitrogen [14,15,16], oxygen [17,18], carbon [19,20], or fluoride [21] nucleophiles to electron-deficient fluoroalkenes, including S_N_2’ reactions [22].

### 2.2. Addition of Nucleophiles Containing Leaving Groups

There are few examples of substitution of halogens in nitroarenes with nucleophiles containing leaving groups, e.g., α-chlorocarbanions, mostly because nucleophilic addition to such arenes proceeds faster from the position occupied by hydrogen. The formed σ^H^ adducts of α-chlorocarbanions usually undergo β-elimination to form products of vicarious nucleophilic substitution (VNS) [23] faster than dissociation and addition at positions occupied by a leaving group (see below, Section 3.2). Nevertheless, there are examples of substitution of halogens, particularly fluorine, in fluoronitrobenzenes by carbanions of α-chloroalkyl sulfones [24,25], alkoxynitriles, sulfenamides, and anion of *t*-butylhydroperoxide (Figure 7) [26]. These reactions have rather limited practical application.

There are also not frequent reports of reactions of such nucleophiles with acyl chlorides or esters of carboxylic acids [27]. Nevertheless, adding anions of hydroperoxides to acyl chlorides is a way of synthesis of acyl peroxides in moderate to good yields (Figure 8) [28].

## 3. Addition of Nucleophiles to π-Electrophiles without Leaving Groups at the Electrophilic Center

### 3.1. Addition of Nucleophiles without Leaving Groups

For many years, the known reactions of nucleophiles with electron-deficient arenes were limited to those proceeding via addition at positions occupied by halogens or other leaving groups. Only recently was it established that adding nucleophiles to nitroaromatic rings also proceeds to rings that do not contain leaving groups. Moreover, addition to the rings containing halogens proceeds faster at positions occupied by hydrogen than halogen [1,2,3,8]. This process was overlooked because the initial fast addition is a reversible process and the initially formed σ^H^ adducts dissociate and slower addition at positions occupied by halogens leading to S_N_Ar taking place (see Figure 1). It should be mentioned that σ^H^ adducts of C, N, O, etc. nucleophiles to highly electrophilic arenes, such as trinitrobenzene, are stable and upon protonation, trinitrocyclohexadiene-type products known as “Meisenheimer complexes”, that are in fact trinitrocyclohexadiene derivatives, are formed (Figure 9) [29].

Carbanions generated from 1,3-dicarbonyl compounds add to such nitroarenes to form upon protonation peculiar bicyclic products in moderate yields (Figure 10) [30]. These important and interesting observations have limited application in organic synthesis.

Anionic σ^H^ adducts of nucleophiles to mononitroarenes are short-lived species; nevertheless, protonation or silylation of such σ^H^ adducts of some carbanions results in the elimination of water to form nitrosoarenes that can be isolated in the form of quinoneoximes [31] or further converted to anthranils (Figure 11) [32].

Silylation of σ^H^ adducts of methinic carbanions followed by elimination of silanol gave substituted nitrosobenzenes (Figure 12) [33].

The scope of these interesting and valuable synthesis processes is rather limited. Short-lived σ^H^ adducts of anilines to mononitroarenes upon protonation also undergo the elimination of water to form nitrosodiphenyl amines (Figure 13) [34,35].

Another important way of further transformation of σ^H^ adducts of C, N, O, and even P nucleophiles to nitroarenes is oxidation by external oxidants to form products of oxidative nucleophilic substitution of hydrogen (ONSH). It was recently shown that this process is of the general character of wide application in organic synthesis. Examples of ONSH in nitroarenes are presented in Figure 14 [33,36,37].

Further, adding nucleophiles to aliphatic π-electrophiles containing carbon-heteroatom double bonds that do not contain leaving groups is energetically favorable; hence, the usual protonation of the adducts provides stable products. This can be exemplified by the addition of cyanide anion to aldehydes and ketones that upon protonation form cyanohydrines or silylation *O*-silylated cyanohydrines (Figure 15) [38,39]. Similarly, the addition of carbanions to aldehydes and ketones followed by protonation gives aldol-type products.

Usually, the protonated adducts—aldols—enter fast further reactions, most often elimination of water to form alkenes (Knoevenagel reaction) [40]. Additionally, adding ammonia and a variety of amines to aldehydes and ketones results in the formation of aminals and subsequently, upon elimination of water, aldimines or enamines (Figure 16).

We would like to stress the similarity of the conversion of the σ^H^ adducts to nitroarenes and to aldehydes and ketones via protonation and the elimination of water. The addition of nucleophiles to highly electron-deficient nitroarenes upon protonation results in formation of relatively stable adducts, similarly to formation of aldols, whereas addition to mononitroarenes followed by protonation and elimination of water gives nitrosoarenes, isolated usually as methylenequinone oximes (Figure 11, Figure 12 and Figure 13) in a process analogous to the Knoevenagel reaction.

Oxidation analogous to that of σ^H^ adducts of nucleophiles to nitroarenes, ONSH, also proceeds with the adducts of nucleophiles to aliphatic π-electrophiles–aldehydes. For instance, aromatic aldehydes treated with potassium permanganate in liquid ammonia form amides in moderate yields (Figure 17) [41].

Formally, the oxidation of protonated adducts of RLi and RMgX to aldehydes, that is alcohols, may be viewed as another example belonging to this category. ONSH reaction also proceeds with electron-deficient alkenes, for example, in quinone or maleimide derivatives [42,43,44,45,46,47] or even nitroalkenes [48] (Figure 18).

ONSH is also a well-established process for the functionalization of two structurally similar classes of π-electrophiles–aromatic *N*-oxides [49,50,51] and aliphatic nitrones (Figure 19) [52,53].

### 3.2. Addition of Nucleophiles Containing Leaving Groups

Of particular interest and value are reactions of aromatic and aliphatic π-electrophiles with nucleophiles that contain leaving groups at the nucleophilic center. Such nucleophiles are exemplified by α-halocarbanions, sulfonium and phosphonium ylides, *t*-butylhydroperoxide anion, substituted amines, etc.

The addition of α-halocarbanions to nitroarenes results in the initial formation of the σ^H^ adducts that subsequently enter base-induced β-elimination of hydrogen halide and protonation to give products of vicarious nucleophilic substitution (VNS). This reaction can proceed provided the σ^H^ adducts exist for a sufficiently long time and the base is present in excess (Figure 20) [23,54].

Similar reaction proceeds with carbanions containing other leaving groups able to be eliminated in the base-induced E2 process. For example, VNS cyanomethylation of nitroarenes in the reaction with aryloxyacetonitriles is widely used to synthesize indoles (Figure 21) [55].

There are a few examples of VNS with sulfonium [56,57,58] and phosphonium ylides (Figure 22) [59,60].

It should be stressed that an identical process of addition—β-elimination proceeds with an anion of *t*-butylhydroperoxide to produce nitrophenols and *N*-anions, generated from some sulfenamides, trimethylhydrazonium iodide, hydroxylamine derivatives, etc. to give nitroanilines (Figure 23) [26,61,62,63].

Thus, VNS is a general process for introducing carbon, nitrogen, and oxygen substituents into aromatic rings.

Interestingly, when delocalization of the negative charge in the σ^H^ adduct of α-halocarbanions or sulfonium ylides to electron-deficient arenes is inefficient, they react further not via β-elimination but via intramolecular substitution to form aziridines or cyclopropanes (Figure 24) [64,65,66,67].

On the other hand, adducts of α-chlorocarbanions and sulfonium ylides to aliphatic electrophiles containing carbonyl groups result in formation of anionic σ adducts that undergo intramolecular 1,3-substitution to form oxiranes (Darzens [68,69] or Corey-Chaykovsky [70,71,72,73] reactions). The reaction of such nucleophiles with imines to form aziridines proceeds similarly (Figure 25) [74,75].

The high affinity of phosphorus to oxygen causes a different behavior of phosphonium ylides. Their anionic adducts to aldehydes and ketones react further via a combination of the *O*-anion with a positively charged phosphorous atom giving oxaphosphetanes, followed by the elimination of phosphine oxide to produce alkenes (Wittig reaction; Figure 26). A similar reaction course leading to alkenes has been reported for adding phosphonium ylides to *N*-sulfonylimines [76,77], whereas simple *N*-arylimines give allenes [78].

The addition of the nucleophiles containing leaving groups to electron-deficient alkenes results in the formation of anionic adducts that are γ-halocarbanions and related intermediates and is followed by intramolecular 1,3-substitution to give cyclopropanes. This is one of the major ways to synthesize substituted cyclopropanes (Figure 27) [71,79,80]. A similar process—addition followed by 1,3-intramolecular substitution reaction—proceeds between the anion of *t*-butylhydroperoxide and some electron-deficient alkenes to give oxiranes, although protonation of the intermediate adducts to form Michael-type products dominates [81]. Similarly, the addition of *N*-nucleophiles containing leaving groups followed by 1,3-intramolecular substitution is an efficient way of synthesis of aziridines [75].

Surprisingly, there are no reports of the possibility of converting adducts of α-chlorocarbanions to aliphatic π-electrophiles containing a carbon-heteroatom double bond via base-induced β-elimination, although such attempts were disclosed in the literature [82]. Motivated by the similarity of aromatic and aliphatic π-nucleophiles reactivity, we have attempted such reactions. β-Elimination in the adducts of α-chlorocarbanions to benzaldehyde does not proceed as it is hampered by the vicinity of the negatively charged oxygen which engages in facile intramolecular substitution to form oxiranes. On the other hand, β-elimination in the adducts of α-chlorocarbanions to electron-deficient imines proceeded satisfactorily (Figure 28) [83].

No examples of β-elimination of HCl from the adducts of α-chlorocarbanions to electron-deficient alkenes were known for a long time. We have shown that such processes, identical to VNS in nitroarenes, proceed under appropriate conditions with electron-deficient alkenes and carbanions containing an aryloxy leaving group or even with α-chlorocarbanions (Figure 29) [42,84,85].

More examples of this kind appeared in recent years, including a reaction of nitroalkanes with NO_2_ acting as a carbanion-stabilizing and leaving group (Figure 30) [86,87]. A reaction of a nitrogen nucleophile bearing a Ph_2_S leaving the group with maleimides and naphthoquinone in good yields has also been reported (Figure 31) [88,89,90].

## 4. Reactions of Specific Nucleophiles

Several kinds of specific nucleophiles also enter analogous reactions with aromatic and aliphatic π-electrophiles. To this category belong 1,3-dipoles that enter 1,3-dipolar cycloaddition both to electron-deficient arenes [91,92] and alkenes [93,94] (Figure 32).

Of substantial interest are masked acyl carbanions, key intermediates in the Stetter reaction [95]. This reaction proceeds via addition of cyanide anion or nucleophilic carbenes to aldehydes and subsequent intramolecular migration of proton to generate masked acyl carbanions that add to active electrophiles, mainly Michael acceptors. Subsequent dissociation gives Michael adducts of acyl carbanions (Figure 33).

Examples of the Stetter reaction with nitroarenes that gave as the final products nitrobenzophenones were reported (Figure 34) [96,97].

It should be mentioned that the Stetter reaction follows the concept of Umpolung introduced by Seebach to invert the polarity of the carbonyl group in aldehydes [98]. Conversion of aldehydes into dithioacetals followed by deprotonation produced masked acyl carbanions. Subsequent reactions with a variety of electrophiles, followed by deacetalization, give products of reactions of acyl carbanions. Reactions of such carbanions with nitroarenes were also reported. A more convenient approach is generating masked acyl carbanions via the conversion of aldehydes into cyanohydrines, followed by protection in the form of acetals via reactions with vinyl ethers (Figure 35). Deprotonation of such protected cyanohydrins gave stable equivalents of acyl carbanions [99,100,101]. Thanks to the stability of all intermediates involved, this way of generating masked acyl carbanions appears to be more versatile than the original Stetter reaction.

## 5. General Comments

Finally, an interesting general question should be addressed. As it was mentioned earlier, nucleophilic addition to *ortho* and *para* halonitrobenzenes proceeds faster at positions occupied by hydrogen than halogens, therefore nucleophilic substitution of hydrogen: VNS, ONSH, etc. proceeds as a rule faster than conventional S_N_Ar of halogens. Based on these observations, it was often considered that halogens in electron-deficient arenes partially protect their positions against nucleophilic addition. It was interesting to see that the same observation was made in rather few experiments with aliphatic π-electrophiles. Thus in 2-chloronaphthoquinone VNS reaction with α-chlorocarbanions proceeds exclusively (Figure 36) [42]. Mayr has shown that nucleophilic addition to chlorobenzoquinones also proceeds faster at positions occupied by hydrogen [102].

We have also found that carbanions add faster at carbon occupied by hydrogen (in position 2) of 1-chlorofumarate and maleate [103]. However, the statement that chlorine decelerates nucleophilic addition at positions it occupies, although supported by many observations, is unjustified simply because competition between addition of nucleophiles at a position occupied by hydrogen and chlorine in *para* or *ortho* chloronitrobenzenes is unfair. Positions occupied by hydrogen are additionally activated by electron withdrawing chlorine substituent in the ring or at the π bond, whereas those occupied by chlorine are not additionally activated. This interesting and important problem should be solved using unbiased models. For instance, we have observed that in competition between nitrobenzene and *p*-chloronitrobenzene for the reaction with a bulky methinic carbanion which does not react in the *ortho* position, the addition of this carbanion in position *para* of nitrobenzene proceeds faster. Similarly, the VNS reaction of chloromalonates with 2-chloronaphthoquinones proceeds faster than with 2,3-dichloronaphthoquinones, also 1,2-dichloromaleate reacts slower than monochloro with carbanion of chloromethyl phenyl sulfone (Figure 36).

These observations support the hypothesis that chlorine substituents indeed decelerate nucleophilic addition at positions they occupy in aromatic and aliphatic π-electrophiles. These interesting observations and hypotheses need further studies.

For instance, there remains an important matter of the relative activity of aldehydes and the corresponding acyl chlorides. We hypothesize that the electrophilic activity of aldehydes is higher, but we are not aware of any experimental evidence. The formation of benzoate of benzaldehyde cyanohydrine when KCN or LiCN is added to a mixture of benzaldehyde and benzoyl chloride is not sufficient [38,104].

## 6. Kinetic vs. Thermodynamic Control

In the electron-deficient aromatic rings, e.g., nitroarenes, there are usually two or even three electrophilic sites able to add nucleophiles, thus the question of kinetic and thermodynamic control is of crucial importance. According to the recently formulated general mechanism of nucleophilic aromatic substitution in nitroarenes, the addition of nucleophiles to nitroarenes proceeds rapidly at positions *ortho* or *para* to the nitro group to form σ^H^ adducts [1,8,105]. The adducts are short-lived species and usually dissociate and slower addition at positions occupied by halogens X followed by the fast departure of halide anion results in the substitution of halogens, thus this process can be considered thermodynamically controlled. Furthermore, when, with proper structure of nucleophiles and conditions, the initially formed σ^H^ adducts are converted into products of substitution of hydrogen, the process can be considered kinetically controlled. Additionally, the orientation of nucleophilic substitution of hydrogen can be kinetically and thermodynamically controlled. For instance, VNS in nitrobenzene with methylenic carbanions under kinetic control—excess of base, low temperature—proceeds mostly in the *ortho* position, whereas under thermodynamic control (r.t., slow addition of the carbanion solution to a solution of nitroarene) *para* substitution dominates. Exemplification of the kinetic vs. thermodynamic control in the reaction of an α-chlorocarbanion with nitroarenes is shown in Figure 37.

These experimental observations are confirmed by calculations of free energies of transition states of the addition and σ adducts [105].

In most aliphatic π-electrophiles, there is one electrophilic center to which the addition of nucleophiles occurs. Nevertheless, there are numerous electrophiles with two such centers, e.g., acrolein, vinyl ketones, acrylates, etc., reactions of which nucleophiles can proceed under kinetic or thermodynamic control (1,2- vs. 1,4-addition) [70].

## 7. Conclusions

We hope that the general concept of this paper—the analogy between reactions of nucleophiles with aliphatic and aromatic π-electrophiles—is convincingly supported by the presented examples and their interpretation. The mechanism of reactions of nucleophiles with both kinds of π-electrophiles are in fact identical; they are all initiated by a nucleophilic addition to the π system, at the position occupied by a leaving group or by hydrogen, with the latter process usually faster. There are several ways of further conversion of the adducts which are common for both aromatic and aliphatic systems. If the π system of the electrophilic partner contains a heteroatom, Knoevenagel-type elimination may also be possible. Importantly, examples of each type of reaction course can be given for both aliphatic and aromatic electrophiles.

Two important conclusions can be drawn from our analysis. First, some types of reactions are underrepresented within either the aliphatic or aromatic electrophiles group, which should inspire the search for new reactions that remain to be discovered. Second, this essay should change the general opinion that reactions of aliphatic electrophiles are more diversified than those of aromatic ones. It is just the opposite, as exemplified by benzoyl chloride, which is capable only of chloride substitution, and chloronitrobenzene which, upon nucleophilic addition, undergoes S_N_Ar, VNS, or two variants of the ONSH reaction [106].

## Data Availability

Data sharing not applicable.

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
