# Peer review of "Analogy of the Reactions of Aromatic and Aliphatic π-Electrophiles with Nucleophiles"

_molecules, 2023, doi:10.3390/molecules28104015_

Round 1

Reviewer 1 Report

Dear Editor,

In this paper, the authors try to perform a review article focused on the mechanistic distinguish of the nucleophilic addition to several aromatic and aliphatic electrophiles.  The presented results are focused to explain the plausible disclose similarity of a great variety of reactions that proceed between nucleophiles and π-electrophiles (aromatic and aliphatic). The combination of aliphatic with aromatic electrophiles confuses the reader. For example, is not appropriate to compare the mechanism of the imine formation with the mechanism of the nitroarenes substitution, et.al. In my opinion, the main aim of such work should be the mechanistic explanation of the aromatic nucleophilic substitution, with and without living groups.

English can be improved.

Author Response

Thank you very much for the Reviewers opinions. It is pity that the Reviewer 1 does not share our opinion that a unified treatment of mechanisms of organic reactions is very useful and well justified. For instance, just using an example given by the Referee – the reaction of aldehydes with amines to form imines and the reaction of nitroarenes with anilines to form nitrosodiarylamines proceed according to an identical mechanism: reversible addition of a nitrogen nucleophile followed by elimination of water. Similarity of the mechanisms is not connected with similarity of products. The mechanistic explanation of the aromatic nucleophilic substitution with and without leaving groups is already published (ref. 5 in the manuscript). 

Reviewer 2 Report

This essay aims to show the similarity between the nucleophiles addition reactions that involve aliphatic and aromatic π-electrophiles. There are versatile examples of such reactions in the literature and I strongly support this attempt to provide a unifying view on them. The manuscript is well written but its scholar representation should be improved by taking care of schemes and in some cases the corresponding text. Some specific points are listed below.

 Abstract (line 11)  Refers to a graphical abstract ("As shown above"). That probably shouldn't be the case and s to be corrected.

I would add X = Hal, AlkO, ArO to Scheme 1

Notations in Scheme 4 (amides, esters, ketons) seem to refer to the reaction products not the starting compounds. Their current position is misleading

For the silylation and silanol elimination steps in Scheme 12 the origin and a fate of protons should be clarified.

Scheme 15 – the product of aldol dehydration should not contain OH group

Scheme 20 – transformation of the σH adducts proceeds formally as an HCl elimination. This can be better reflected on the scheme, e.g. as a real form in which H and Cl are removed by excess base.

The mechanism of reactions on Scheme 32 deserves to be shown/described

Scheme 33 – phenyl group is missing in the second row

Scheme 35 – the first reaction step - synthesis of cyanohydrin ether – is not as simple as protonation of parent cyanohydrin, please correct

lines 464-465 "…the recently formulated general mechanism of nucleophilic aromatic substitution in nitroarenes…" requires a proper reference

Author Response

 Abstract (line 11)  Refers to a graphical abstract ("As shown above"). That probably shouldn't be the case and s to be corrected. done

I would add X = Hal, AlkO, ArO to Scheme 1 done

Notations in Scheme 4 (amides, esters, ketons) seem to refer to the reaction products not the starting compounds. Their current position is misleading done

For the silylation and silanol elimination steps in Scheme 12 the origin and a fate of protons should be clarified. done

Scheme 15 – the product of aldol dehydration should not contain OH group done

Scheme 20 – transformation of the σH adducts proceeds formally as an HCl elimination. This can be better reflected on the scheme, e.g. as a real form in which H and Cl are removed by excess base. done

The mechanism of reactions on Scheme 32 deserves to be shown/described done

Scheme 33 – phenyl group is missing in the second row done

Scheme 35 – the first reaction step - synthesis of cyanohydrin ether – is not as simple as protonation of parent cyanohydrin, please correct done

lines 464-465 "…the recently formulated general mechanism of nucleophilic aromatic substitution in nitroarenes…" requires a proper reference done

Reviewer 3 Report

In this outstanding tutorial review (essay), the authors outline nucleophilic substitutions at sp2-centers. Their work clearly demonstrates an analogy between the reactions of electron-deficient aromatic compounds and electron-poor alkenes (and in some cases carbonyls/acyl groups). Both undergo an addition step preferably at the hydrogen site, but the fate of the reaction is determined by subsequent steps. I recommend publication.

The reviewer holds an opinion that the terms aromatic and aliphatic are not antonyms. The term "aliphatic electrophile" is rare and most commonly used for actual aliphatic compounds such as iodoethane. Electron-poor olefins and naphthoquinones do not fit well in this category.

The yields are not discussed, and they are rarely shown in the schemes. It would be helpful to include the yields. It is recommended to provide yields, yield ranges, or yield comments such as "high/low yield" at least in the schemes for the less obvious chemistry.

Other specific and minor comments:

Ln 33: The authors are referencing the Abstract figure, not the Abstract itself.

Ln 252: In the first row, check the oxidant source.

Ln 343: In the Scheme, explain Q+Cl-.

Ln 394: In the Scheme, the Ph group was not drawn in the first structure, second row.

Ln 481: In the Table, the percentages indicate product distribution; please label accordingly.

Ln 40: An article is missing.

Ln 49: Correct hyphenation.

Ln 208: Consider using parentheses.

Ln 237: Schemes.

Lns 359, 406, and 453: Improve the writing.

Author Response

In this outstanding tutorial review (essay), the authors outline nucleophilic substitutions at sp2-centers. Their work clearly demonstrates an analogy between the reactions of electron-deficient aromatic compounds and electron-poor alkenes (and in some cases carbonyls/acyl groups). Both undergo an addition step preferably at the hydrogen site, but the fate of the reaction is determined by subsequent steps. I recommend publication.

The reviewer holds an opinion that the terms aromatic and aliphatic are not antonyms. The term "aliphatic electrophile" is rare and most commonly used for actual aliphatic compounds such as iodoethane. Electron-poor olefins and naphthoquinones do not fit well in this category.

We understand the above reservations of the Referee 2 concerning terminology. On the other hand, for the purposes of our text it would be difficult to find a different, short and general label for electrophiles that are not aromatic. Moreover, for example simple alkenes are considered as aliphatic compounds in most textbooks, so we believe that it is also acceptible to call “aliphatic” their derivatives such as unsaturated ketones, esters, etc.

The yields are not discussed, and they are rarely shown in the schemes. It would be helpful to include the yields. It is recommended to provide yields, yield ranges, or yield comments such as "high/low yield" at least in the schemes for the less obvious chemistry. Done

Other specific and minor comments:

Ln 33: The authors are referencing the Abstract figure, not the Abstract itself. Corrected

Ln 252: In the first row, check the oxidant source. It is O3 as drawn.

Ln 343: In the Scheme, explain Q+Cl-. done

Ln 394: In the Scheme, the Ph group was not drawn in the first structure, second row. done

Ln 481: In the Table, the percentages indicate product distribution; please label accordingly. done

Comments on the Quality of English Language

Ln 40: An article is missing. Corrected

Ln 49: Correct hyphenation. done

Ln 208: Consider using parentheses. done

Ln 237: Schemes. done

Lns 359, 406, and 453: Improve the writing. done